# TEG^®^, Microclot and Platelet Mapping for Guiding Early Management of Severe COVID-19 Coagulopathy

**DOI:** 10.3390/jcm10225381

**Published:** 2021-11-18

**Authors:** Gert Jacobus Laubscher, Petrus Johannes Lourens, Chantelle Venter, Douglas B Kell, Etheresia Pretorius

**Affiliations:** 1Mediclinic Stellenbosch, Stellenbosch 7600, South Africa; laubscher911@gmail.com (G.J.L.); wodie22@icloud.com (P.J.L.); 2Department of Physiological Sciences, Faculty of Science, Stellenbosch University, Stellenbosch 7602, South Africa; chantellev@sun.ac.za; 3Department of Biochemistry and Systems Biology, Institute of Systems, Molecular and Integrative Biology, Faculty of Health and Life Sciences, University of Liverpool, Liverpool L69 7ZB, UK; 4The Novo Nordisk Foundation Centre for Biosustainability, Technical University of Denmark, 2800 Kgs. Lyngby, Denmark

**Keywords:** COVID-19, consumptive coagulopathy, platelets, blood clotting, fibrinolysis, von Willebrand factor

## Abstract

An important component of severe COVID-19 disease is virus-induced endothelilitis. This leads to disruption of normal endothelial function, initiating a state of failing normal clotting physiology. Massively increased levels of von Willebrand Factor (VWF) lead to overwhelming platelet activation, as well as activation of the enzymatic (intrinsic) clotting pathway. In addition, there is an impaired fibrinolysis, caused by, amongst others, increased levels of alpha-(2) antiplasmin. The end result is hypercoagulation (proven by thromboelastography^®^ (TEG^®^)) and reduced fibrinolysis, inevitably leading to a difficult-to-overcome hypercoagulated physiological state. Platelets in circulation also plays a significant role in clot formation, but they themselves may also drive hypercoagulation when they are overactivated due to the interactions of their receptors with the endothelium, immune cells or circulating inflammatory molecules. From the literature it is clear that the role of platelets in severely ill COVID-19 patients has been markedly underestimated or even ignored. We here highlight the value of early management of severe COVID-19 coagulopathy as guided by TEG^®^, microclot and platelet mapping. We also argue that the failure of clinical trials, where the efficacy of prophylactic versus therapeutic clexane (low molecular weight heparin (LMWH)) were not always successful, which may be because the significant role of platelet activation was not taken into account during the planning of the trial. We conclude that, because of the overwhelming alteration of clotting, the outcome of any trial evaluating an any single anticoagulant, including thrombolytic, would be negative. Here we suggest the use of the degree of platelet dysfunction and presence of microclots in circulation, together with TEG^®^, might be used as a guideline for disease severity. A multi-pronged approach, guided by TEG^®^ and platelet mapping, would be required to maintain normal clotting physiology in severe COVID-19 disease.

## 1. Introduction

The coronavirus disease 2019 (COVID-19) caused by the SARS-CoV-2 virus has led to a worldwide, sudden and substantial increase in hospitalizations for pneumonia with multi-organ problems [1,2,3]. Severe cases of COVID-19 are almost inevitably accompanied by respiratory failure and hypoxia, and treatment includes best practices for supportive management of acute hypoxic respiratory failure [1]. Although, initially thought to be a typical viral pneumonia with acute disseminated intravascular coagulopathy (DIC), it has now been accepted that COVID-19 is primarily an endothelial disease [4] and vascular disease [5]. Endotheliopathy plays a significant role in the severity of COVID-19 [4,5,6,7,8,9,10,11]. Endotheliopathy is also known to be significantly linked to coagulopathies, as it activates microthrombotic pathways and initiates microthrombogenesis, leading to endotheliopathy-associated intravascular microthrombi [12].

Approximately 20% of COVID-19 patients rapidly progress to severe illness characterized by atypical interstitial bilateral pneumonia, acute respiratory distress syndrome and multi-organ dysfunction [9]. It was also shown that one-third of patients hospitalized due to severe COVID-19 develop macrovascular thrombotic complications, including venous thromboembolism, stroke and myocardial injury/infarction [13]. Patients also suffer from right ventricular dilation of the heart [14,15,16]. Autopsies from COVID-19 patients also revealed multi-organ damage pattern consistent with microvascular injury [11,13]. Furthermore, autopsy results have also confirmed microthrombi throughout the lung [17]. In 2020, Ackermann and co-workers reported that histologic analysis of pulmonary vessels in patients with COVID-19 showed widespread thrombosis with microangiopathy [8]. The authors also found that alveolar capillary microthrombi were nine times as prevalent in patients with COVID-19 as in patients with influenza (*p* < 0.001) [8]. In autopsy samples of lungs from patients with COVID-19, the amount of new vessel growth—predominantly through a mechanism of intussusceptive angiogenesis (i.e., splitting of an existing vessel where the capillary wall extends into the lumen of an existing vessel)—was 2.7 times as high as that in the lungs from patients with influenza (*p* < 0.001) [8]. Middleton and co-workers showed the presence of neutrophil extracellular traps (NETs) in lung autopsy results, and suggested that these may be the cause of immune-thrombosis and may, in part, explain the prothrombotic clinical presentations in COVID-19 [18]. Menter and co-workers also showed autopsy findings from 21 COVID-19 patients, and reported that the primary cause of death was respiratory failure, with exudative diffuse alveolar damage and massive capillary congestion, often accompanied by microthrombi despite anticoagulation therapy [19]. A possible reason might be because the extent of systemic hypercoagulation was too significant for the medication to have a substantial enough effect. Hypercoagulability, resulting in a profoundly prothrombotic state, is a distinct feature of the early stages of COVID-19 and is accentuated by a high incidence of fibrinolysis shutdown [20,21,22,23,24,25,26,27,28,29,30]. In fact, Chaudhary and co-workers in 2021 reported that all current studies support COVID-19 as a hypercoagulable and hypofibrinolytic state in the ICU setting [31]. In summary, all the current studies support COVID-19 as a hypercoagulable and hypofibrinolytic state in the ICU setting.

Healthy soluble fibrinogen is referred to in this paper as fibrinogen and healthy fibrin nets that form during the normal physiological processes are referred to as fibrin. Circulating inflammatory molecules, (including inflammagens from viruses and bacteria), may bind to fibrinogen, causing some of the proteins to polymerize into microclots. Hence the use of the term fibrin(ogen) is referred to both soluble and polymerized fibrin/fibrinogen, perceived as unhealthy. Fibrin(ogen) might also have an anomalous (or amyloid) nature [32,33,34]. We have also recently shown that in COVID-19, the healthy fibrinogen changes to an amyloid form (fibrin(ogen)), and that platelets are hyperactivated, and that they may form complexes with erythrocytes [21,22]. Proteomics also revealed that there are significantly dysregulated clotting proteins in microclots, including significant increases in the molecule *a*-(2)-antiplasmin, which prevents fibrinolysis [23].

### 1.1. Disseminated Intravascular Coagulopathy (DIC) and COVID-19: An Uncommon Phenomenon?

From the vast literature in COVID-19 and clotting, it is now well-accepted that coagulopathies occur in the majority of patients who die from COVID-19 [35,36], and that DIC severe bleeding events are uncommon in COVID-19 patients [11]. However, COVID-19 can be complicated by DIC, which has a strongly prothrombotic character with a high risk of venous thromboembolism [26]. It was also noted that sepsis-induced coagulopathy and the International Society of Thrombosis and Hemostasis (ISTH) overt DIC scores (assessed in 12 patients who survived and eight patients who died), increased over time in patients who died. The onset of sepsis-induced coagulopathy was typically before overt DIC [37]. A report from Wuhan, China, indicated that 71% of 183 individuals who died of COVID-19 met criteria for DIC [1,35]. Spiezia and co-workers argued in most COVID-19 patient’s high D-dimer levels are associated with a worse prognosis [38]. The authors also suggest that COVID-19 patients with acute respiratory failure represent the consequence of severe hypercoagulability, that when left untreated results in consumptive coagulopathy (end-stage (DIC)) and that excessive fibrin formation and polymerization may predispose to thrombosis and correlate with a worse outcome [38]. Consumptive coagulopathy is characterized by abnormally increased activation of procoagulant pathways. This results in intravascular fibrin(ogen) deposition and decreased levels of hemostatic components, including platelets, soluble fibrinogen, and other clotting factors. DIC results in bleeding and intravascular thrombus formation that can lead to tissue hypoxia, multiorgan dysfunction, and death [39]. It is therefore worth noting that DIC is a thrombotic coagulopathy that eventually leads to bleeding. However, in COVID-19, DIC (as defined by the ISTH or other bodies) is an uncommon phenomenon.

### 1.2. The Progression of the Disease, If Untreated Is a Two-Phase ‘Rollercoaster’ of Events, Characterized by Thrombotic Pathology Followed by Bleeding or Thrombocytopenia Pathologies

In 2020, Aigner and co-workers suggested that there could be four strategies in approaching COVID-19 treatment [40]: (1) antiviral treatments to limit the entry of the virus into the cell, (2) anti-inflammatory treatment to reduce the impact of COVID-19 associated inflammation and cytokine storm, (3) treatment using cardiovascular medication to reduce COVID-19 associated thrombosis and vascular damage, and (4) treatment to reduce the COVID-19 associated lung injury. As we have learnt more about the disease, we now know that the cytokine storm (followed by bleeding) is only a concern if the disease is left untreated. In 2020 we presented evidence that COVID-19 can be seen as a two-phase ‘rollercoaster’ of events, characterized by serial (i) thrombotic and (ii) bleeding or thrombocytopenia pathologies [41]. These substantial vascular events are significant accompaniments to ARDS and lung complications and both vascular events are seen in COVID-19 patients [42,43,44,45,46,47,48]. Clearly these coagulopathies seem to represent polar opposites, and it might be seen as odd if both are said to accompany COVID-19 pathology; the resolution of the apparent paradox is that these coagulopathies can be differentiated in time. Figure 1 shows the fine balance during COVID-19, between these biomarkers and the development of initial hyperclotting and thrombosis that can be followed by a consumptive coagulopathy, thrombocytopenia and bleeding; the latter is followed by the cytokine storm (at the end stage of the disease) [41]. Depending on the direction (i.e, increases or decreases), dysregulation of fibrin(ogen), D-dimer, von Willebrand Factor (VWF) and P-selectin may result in either hypercoagulation or excessive bleeding and thrombocytopenia (hypocoagulation). We suggested that patients need to be treated early in the disease progression, when hypercoagulation is clinically diagnosed (discussed later in the treatment protocol). Early in the hypercoagulation phase of the disease, high levels of VWF, P-selectin and fibrinogen are present, but that there are still normal or slightly increased levels of D-dimer. If the disease is left to progress until the patient presents with VWF and fibrin(ogen) depletion, and with high D-dimer levels (and even higher P-selectin levels), it will be indicative of a poor prognosis, an imminent cytokine storm and ultimately death. This rollercoaster disease progression is a continuum and the progression of disease has no specific tipping point (Figure 1). In a 2020 JAMA editorial, the question was also asked whether the cytokine storm could be seen as significantly relevant in COVID-19, and it was referred to as “tempest in a teapot” [49]. The basis for this conclusion was that the presence of elevated circulating mediators in the claimed cytokine storm are likely to reflect endothelial dysfunction and systemic inflammation leading to fever, tachycardia, tachypnea, and hypotension [49], rather than the more immediately lethal ARDS. The JAMA editorial concluded by suggesting that incorporating the cytokine storm may only further increase uncertainty about how best to manage this heterogeneous population of patients [49]. Our rollercoaster diagram (Figure 1) also notes that increased levels of inflammatory cytokines will already start early in the disease [41].

During the progression of COVID-19, the circulating biomarkers P-selectin, VWF, fibrin(ogen) and D-dimer may either be within healthy levels, upregulated or eventually depleted [21,22]. In COVID-19 patients, dysregulation, has been noted in each of them and this may lead to the extensive endotheliopathy noted in COVID-19 patients [7,8,50] (and see Table 1). Endotheliopathy could give rise to hypercoagulation by alteration in the levels of different factors such as VWF [10]. Fibrinogen concentration is also a static measure and does not provide information about functionality [31]. D-dimer might also have low specificity and the elevated levels may be related with other conditions. This phenomenon was discussed by Kabrhel and co-workers in 2010, where the authors noted that many factors are associated with a positive D-dimer test, including age, active malignancy and conditions such as lupus and rheumatoid arthritis [31].

With the above-mentioned pathologies in mind, we here argue that the early treating of COVID-19 disease as a vascular and endothelial disease with severe platelet dysfunction and microclot formation, early on in the disease, provides a framework for a rational treatment strategy. We review literature on the use of point-of-care equipment such as the thromboelastography^®^ (TEG^®^) and the PFA200, to manage patient with COVID-19. We also suggest the use of the degree of platelet dysfunction and presence of microclots in circulation, using fluorescence microscopy, as a guideline for disease severity.

### 1.3. A Place for Existing Point-Of Care Techniques to Guide COVID-19 Treatment

Thromboelastometry (TEM), also known as rotational thromboelastography (ROTEG) or rotational thromboelastometry (ROTEM), is an established viscoelastic method for hemostasis testing. It is a modification of traditional TEG^®^. These techniques are crucial point-of-care techniques that we suggest might be used to guide the treatment of COVID-19 patients. Spiezia and colleagues also noted that COVID-19 patients with acute respiratory failure present with severe hypercoagulability due to hyperfibrinogenemia, resulting in increased polymerized fibrin cross-linking formation that may predispose the patient to thrombosis. Spiezia and co-workers also concluded that thromboelastometry is an important point-of-care test in COVID-19, as it has the advantage of providing a global assessment of whole blood’s ability to clot. On the other hand, it is not able to evaluate the contribution to clot formation of each element (including endothelium, platelets, and clotting factors). In 2020, Wright and co-workers also discussed the use of clot lysis at 30 min (LY30) on the TEG^®^ as point-of-care analysis method [46]. The LY30 parameter (measured in %) is recorded at 30 min after the point where the maximum amplitude (MA) of the clot is reached (see Figure 2). LY30 of 3% or greater defines clinically relevant hyperfibrinolysis [61]. The TEG^®^ results, particularly an increased MA and G-score (that both measures maximal clot strength) is used to predict thromboembolic events and a poor outcome in critically ill patients with COVID-19 [46]. MA is of great significance as it represents clot size (see Figure 2), as determined by platelet number and function, as well as polymerized fibrin cross-linking to form a stable clot. 

Recently, various papers have shown the significance of TEG^®^, and the levels of coagulopathy in COVID-19 (an important measurement value of the TEG^®^) in managing COVID-19 patients is also getting more traction [62,63,64,65]. Hranjec and co-workers in 2020 also noted that TEG^®^ with platelet mapping, better characterizes the spectrum of COVID-19 coagulation-related abnormalities and may guide more tailored, patient-specific therapies these patients [64]. Another important test is the PFA-200 platelet test. This test may be seen as a cross between bleeding time and quick aggregation testing. See Table 2 for the various parameters for the TEG^®^ and PFA-200. 

An important consideration is that TEG^®^ can be used to study the clotting parameters of both whole blood (WB) and platelet poor plasma (PPP). Whole blood TEG^®^ gives information on the clotting potential affected by the presence of both platelets and fibrinogen, while PPP TEG^®^ only presents evidence of the clotting potential of the plasma proteins [67,68,69,70]. Reasons for a hypercoagulable TEG^®^ trace when using PPP, may be indicative of the presence of dysregulated inflammatory biomarkers, including P-selectin, inflammatory cytokines and increased levels of fibrin(ogen) [71,72,73,74,75,76]. 

Typical laboratory pathology tests are usually conducted on plasma (after the cellular material has been removed that includes the platelets). In these tests the platelets are literally discarded and therefore ignored. Well-known coagulation tests such as the prothrombin time (PT) and partial thromboplastin time (PTT), has been shown not to give a true reflection of the hypercoagulable state in acute COVID patients [77], as these parameters ignore other components of the coagulation such as the platelet function and fibrinolysis [31]. On the other hand, whole blood viscoelastic analysis can be rapidly performed by TEG or ROTEM, as these techniques measure the whole blood capability might be kept to the minimum by standardized protocols to investigate the utility of TEG/ROTEM in assessing risk for thrombosis and bleeding [78].

We have also found that during the presence of systemic inflammation, reflected in an increased presence of inflammagens, the biochemistry of the fibrin(ogen) molecule changes its folding characteristics considerably (Figure 3), to produce amyloid forms. We could visualize these changes using fluorescence markers [32,57,79,80,81]. The fluorescence markers we have used to show these structural changes in the fibrin(ogen) biochemistry included thioflavin T (ThT) and various Amytracker dyes. These fluorescence markers are typically used to show amyloid changes to proteins [32,57,79,80,81], suggesting the misfolding seen in fibrin(ogen) during the presence of inflammagens in the blood, could also be described as amyloid. ThT binds to open hydrophobic regions on damaged protein [57,81]. We showed, that when healthy fibrinogen is exposed to increased levels of inflammatory biomarkers and bacterial (viral) inflammagens, PPP TEG^®^ traces was significantly hypercoagulable [79,80,81,82]. Figure 3A shows the uncoiling of the fibrin(ogen) molecule where it caused plasma and WB to become hypercoagulable. Figure 3B,C shows scanning electron micrographs of representative examples of a healthy clot and a clot from a Type 2 Diabetes (T2DM) individual (taken from [76]). 

The value of using both the TEG^®^ and fluorescence markers was again seen in our recent studies on the effects that COVID-19 has on the coagulation system, where an increased clot strength and anomalous microclot formation were observed [21,22]. 

## 2. The Early Treatment of Patients with Anticoagulation Medication 

Early on in the disease, it was suggested that best practices for supportive management of acute hypoxic respiratory failure and (ARDS can be followed [1,83]. In 2020, the American Thoracic Society-led international task force has also released a guidance document to help clinicians manage COVID-19. The new guidance—“COVID-19: Interim Guidance on Management Pending Empirical Evidence”—is published as an open access document on the American Thoracic Society’s website (https://www.thoracic.org/covid/covid-19-guidance.pdf (accessed on 15 October 2021)) [84]. These guidelines in broad terms suggest the use of ventilation in patients who have refractory hypoxemia and ARDS, to consider extracorporeal membrane oxygenation in patients who have refractory hypoxemia, COVID-19 pneumonia (i.e., ARDS), and have failed prone ventilation. Most significant is that the guidelines recognize and state that: “We believe that in urgent situations like a pandemic, we can learn while treating by collecting real-world data”. Evidence-based guideline initiatives have also been established by many countries and professional societies, e.g., in South Africa, the South African Society of Anaesthesiologists (SASA) has detailed guidelines on their website (https://sasacovid19.com/ (accessed on 15 October 2021)). In addition, guidelines are updated on a very regular basis by the National Institutes of Health (https://www.covid19treatmentguidelines.nih.gov/ (accessed on 15 October 2021)). Potential therapies that address vascular system dysfunction and its sequelae may have an important role in treating patients with COVID-19 and its long-lasting effects [5]. Heparin has also been found in some circumstances to be a helpful treatment for COVID-19 [85,86]. Anticoagulant therapy mainly with low molecular weight heparin might associated with better prognosis in severe COVID-19 patients, meeting sepsis-induced coagulopathy criteria, or with markedly elevated D-dimer [87]. Heparin assists in the prevention of thrombotic events, by interacting with anti-thrombin III. It was also recently shown, in a study of 449 patients with severe COVID-19, that anticoagulant therapy, mainly with low molecular weight heparin, appeared to be associated with lower mortality in the subpopulation meeting sepsis-induced coagulopathy criteria or with markedly elevated D-dimer [26,87]. In this study 99 of the patients received heparin (mainly low molecular weight heparin) for 7 days or longer. One of the essential conclusions were that heparin treatment appears to be associated with better prognosis in severe COVID-19 patients with coagulopathy. We suggest that the timeline of the rollercoaster disease progression can be hours and it is a continuum rather than a clear event or “flip” between hypercoagulation and bleeding. Timing of treatment is therefore essential. If the disease is left unabated, VWF and fibrin(ogen) depletion, and significantly increased levels of D-dimer and P-selectin will progress on a continuum [41]. 

### Anticoagulation Trails

Here we argue that an early and aggressive treatment with a multipronged approach covering the enzymatic pathway, platelets and lysis of microclots, are key in the treatment of COVID-19. Recently, there were two trials we wish to focus on, that studied the effects of anticoagulation [58,88]. 

In an open-label, adaptive, multiplatform, controlled trial, published in New England Journal of Medicine, Lawler and co-workers randomly assigned patients who were hospitalized with COVID-19 and who were not critically ill to receive pragmatically defined regimens of either therapeutic-dose anticoagulation with heparin or usual-care pharmacologic thromboprophylaxis [88]. The findings suggested that in noncritically ill patients with COVID-19, an initial strategy of therapeutic-dose anticoagulation with heparin increased the probability of survival to hospital discharge with reduced use of cardiovascular or respiratory organ support as compared with usual-care thromboprophylaxis. In an editorial, referring to this paper, Ten Cante [89] reiterates the suggestion that intermediate to full anticoagulation is indicated in patients with moderate disease. 

In another open-label, multicenter, randomized, controlled trial, published in the Lancet in 2021, researchers investigated therapeutic versus prophylactic anticoagulation for patients admitted to hospital with COVID-19 and elevated D-dimer concentration [58]. In this trial, 3331 patients were screened and 615 were randomly allocated (311 (50%) to the therapeutic anticoagulation group and 304 (50%) to the prophylactic anticoagulation group. In the study, 576 (94%) were clinically stable and 39 (6%) were clinically unstable. The primary efficacy outcome was not different between patients assigned therapeutic or prophylactic anticoagulation, with 28,899 (34.8%) similar results in the therapeutic group and 34,288 (41.3%) in the prophylactic group (win ratio 0.86 (95% CI 0.59–1.22), *p* = 0.40). Consistent results were seen in clinically stable and clinically unstable patients. The primary safety outcome of major or clinically relevant non-major bleeding occurred in 26 (8%) patients assigned therapeutic anticoagulation and seven (2%) assigned prophylactic anticoagulation (relative risk 3.64 (95% CI 1.61–8.27), *p* = 0.0010).

The treatment regime that was followed:Therapeutic anticoagulation was in-hospital oral rivaroxaban (20 mg or 15 mg daily) for stable patients, or initial subcutaneous enoxaparin (1 mg/kg twice per day);or intravenous unfractionated heparin (to achieve a 0.3–0.7 IU/mL anti-Xa concentration) for clinically unstable patients, followed by rivaroxaban to day 30;Prophylactic anticoagulation was standard in-hospital enoxaparin or unfractionated heparin.

The results from the trial suggest that in-hospital therapeutic anticoagulation with rivaroxaban or enoxaparin followed by rivaroxaban to day 30 did not improve clinical outcomes and increased bleeding compared with prophylactic anticoagulation. The conclusion of data from the trial was therefore that the use of therapeutic-dose rivaroxaban, and other direct oral anticoagulants, might be avoided in these patients in the absence of an evidence-based indication for oral anticoagulation.

In 2020, Viecca and co-workers reported on a single center, investigator initiated, proof of concept, case control, phase IIb study (NCT04368377) conducted in Italy [90]. The study explored the effects anti-platelet therapy on arterial oxygenation and clinical outcomes in patients with severe COVID-19 with hypercoagulability. Patients received 25 μg/Kg/body weight tirofiban as bolus infusion, followed by a continuous infusion of 0.15 μg/Kg/body weight per minute for 48 h. Before tirofiban, patients received acetylsalicylic acid 250 mg infusion and oral clopidogrel 300 mg; both were continued at a dose of 75 mg daily for 30 days. Fondaparinux 2.5 mg/day sub-cutaneous was given for the duration of the hospital stay. The investigators found that antiplatelet therapy might be effective in improving the ventilation/perfusion ratio in COVID-19 patients with severe respiratory failure and that the therapy prevent clot formation in lung capillary vessels [90].

A recent publication discussed the outcomes of the largest observational study to date, of prehospital antiplatelet therapy in patients with COVID-19, where a significantly lower in-hospital mortality was seen in the group that received antiplatelet therapy. The propensity score-matched cohort of 17,347 patients comprised of 6781 and 10,566 patients in the antiplatelet and non-antiplatelet therapy groups, respectively. In-hospital mortality was significantly lower in patients receiving prehospital antiplatelet therapy (18.9% vs. 21.5%, *p* < 0.001), resulting in a 2.6% absolute reduction in mortality (HR: 0.81, 95% CI: 0.76–0.87, *p* < 0.005).

Our analysis of the conclusions of the above-mentioned trials are as follows: To have a chance at supporting clotting physiology, the enzymatic clotting pathway, as well as platelet activation need to be aggressively controlled and guided by TEG^®^ and platelet functional assays. The aim of treatment is to restore/maintain normal clotting physiology and not aiming for a hypocoagulable state. By conducting random controlled trials and severe COVID-19 patients and trying to prove the efficacy of prophylactic vs therapeutic clexane (low molecular weight heparin treatment (LMWH)) (or any single agent for that matter, affecting clotting) would be futile because the important part that the platelet plays, is ignored. The trial would have a predictable negative outcome for both doses of LMWH and one might come to the (incorrect) conclusion that there is no place for anticoagulation in these patients.

There are also other trials including a Montelukast trial for COVID-19 (https://clinicaltrials.gov/ct2/show/NCT04714515 (accessed on 15 October 2021)). The aim of this trial is to make a therapeutic comparison and effectiveness of Hydroxychloroquine and Montelukast in COVID-19 patients in addition to the standard of care. It was also found by Khan and co-workers in 2021, that hospitalized COVID-19 patients treated with montelukast had fewer events of clinical deterioration, indicating that this treatment may have clinical activity [91]. While this retrospective study highlights a potential pathway for COVID-19 treatment. 

It was suggested that montelukast can directly impact on COVID-19 [92], by having an anti-viral effect, or by suppression of heightened cytokine release in response to the virus [93,94,95,96]. Montelukast can block Cysteinyl-leukotriene in different organs or indirectly through inhibition of the NF-κB signaling pathway [97]. 

As Montelukast primarily acts on the leukotriene pathway, and might only be useful at the end-stage of the disease when the cytokine storm is a major concern. According to our understanding of the disease, i.e., treating the hypercoagulable state early on, using Montelukast (a leukotriene re-uptake antagonist) at this stage would have no effect on outcome, because the molecule has no effect on COVID-19 coagulopathy. 

## 3. Fluorescence Microscopy Can Be Used to Provide a Marker of Microclot Formation and Platelet Hyperactivation 

Currently there is no effective pathology laboratory diagnosis of acute COVID-19, based on the presence of microclots or level of platelet hyperactivation. However, we developed a fluorescence microscopy-based grading system for both microclot presence as well as platelet hyperactivation. Although specialized fluorescence microscopy is needed, both the methods are relatively easy and greatly cost-effective if a fluorescence microscope is available for the analysis. 

### 3.1. Methods Used to Analyse Microclots in Platelet Poor Plasma

To view anomalous clotting of fibrin(ogen) and plasma proteins, in platelet poor plasma (PPP), blood is drawn in citrate tubes, and PPP is collected after a centrifugation step of 15 min at 3000 RPM. This method was previously discussed in various of our papers [21,22,23,57]. To view microclots, thioflavin T (ThT) (exposure concentration: 5 μM) (Sigma-Aldrich, St. Louis, MO, USA) is added to PPP and incubated for 30 min. After placing a 3 µL drop of the sample on a microscope slide, the sample is viewed with a fluorescence microscope. In our case, we used the Zeiss Axio Observer 7 fluorescence microscope with a Plan-Apochromat 63×/1.4 Oil DIC M27 objective (Carl Zeiss Microscopy, Munich, Germany) using the excitation wavelength of 450 nm to 488 nm and emission from 499 nm to 529 nm. 

### 3.2. Methods Used to Prepare Platelets

The analysis of platelet activation might be extremely difficult, as platelets are easily activated. Flow cytometry and platelet aggregation tests are therefore extremely tedious and difficult procedures, even for pathology laboratories. We therefore developed a method where we use the hematocrit of the citrated blood sample to study platelet activation in patients [21,22,23]. In patients with COVID-19, platelets are extremely fragile and hyperactivate easily [11,98]. The whole blood centrifugation step to prepare the hematocrit will therefore easily cause further platelet activation. However, in healthy individuals, this step will have limited influence on platelets, only triggering slight pseudopodia formation. This method could therefore also be seen as a method to show platelet activation under the stress induced by a centrifugation step [22,23]. 

After preparing a hematocrit sample as described previously and removing the PPP, we add two fluorescent antibodies, to CD62P (platelet surface P-selectin) (IM1759U, Beckman Coulter, Brea, CA, USA) and to PAC-1 (activated GP IIb/IIIa) (340507, BD Biosciences, San Jose, CA, USA) to the hematocrit [99] CD62P/P-selectin is released from the cellular granules during platelet activation and then moves to the surface of the platelet membrane. The antibody PAC-1 detects the neoepitope of active GPIIb/IIIa. PAC-1 antibody binding is correlated with platelet activation. After the samples are incubated at room temperature for 30 min it can also be viewed using a fluorescence microscope. We used the Zeiss Axio Observer 7 fluorescence microscope with a Plan-Apochromat 63×/1.4 Oil DIC M27 objective (Carl Zeiss Microscopy, Munich, Germany), using the excitation wavelength 406 to 440 nm and the emission at 546 to 564 nm for the PAC-1 marker, while the excitation for the CD62P was set at 494–528 nm and the emission 618 to 756 nm [22,23].

### 3.3. A Grading System for Plasma Microclot Formation and Platelet Activation, Spreading and Clumping

Figure 4, Figure 5 and Figure 6 shows plates of the various stages of microclot formation and also platelet activation, spreading and clumping, respectively. We suggest a grading system for the use of both microclot presence and platelet activation, as shown in Table 3 and Table 4. The last row of micrographs in Figure 4, show microclots using bright-field microscopy, which is a standard light microscopy method that can be used by pathology laboratory. After combining the scores of both the platelet criteria and PPP criteria, Table 5 shows an example of a combined scoring criterion to produce an overall result score for a patient of both microclot and platelet activation. This scoring result may be used as a guide for the level of pathological clotting present in a patient. Together with thromboelastography analysis, this grading could guide clinical practice.

A prognostic indicator score is also suggested to determine risk of developing severe disease (see Table 6). This score indicator system could allow the clinician to allocate points for various parameters, including age, effort intolerance, hypoxemia, O_2_ saturation, chest Z-ray and/or CT scan carotid intima-media thickness, and other co-morbidities. In addition, a scoring based on parameters from the point-of-care TEG is also suggested.

## 4. Conclusions

The identification and development of new diagnostic methods, based on emerging research and clinical evidence that may assist the identification of novel therapeutic candidates or treatment regimens is crucial. We suggest here that an approach of early and close monitoring of clinical parameters of clotting, including TEG^®^ parameters, microclot and platelet mapping, are crucial in the successful management of COVID-19 patients. However, it requires clinicians and researchers to be flexible with a willingness to embrace new diagnostic methods and treatment protocols that might deviate slightly from the recognized protocols. With this statement we most certainly do not imply that clinicians and researchers might lower the bar for standards of evidence. However, during this pandemic with the rapidly changing environment, traditional rules may not apply [100]. As the coagulopathy changes over time (and this timeframe can be within hours), therapy might be guided by clinical parameters, including TEG^®^ parameters, levels of healthy fibrinogen, VWF, as well as D-dimer. Beta-thromboglobulin and platelet factor 4 may also be used for detecting increased platelet activation in vivo [101,102]. In 1981, Kaplin already pointed out that the measurement of plasma levels of beta-thromboglobulin and platelet factor 4 can be is useful [103]. In addition, in future the roles of erythrocytes and blood rheology might also be further investigated [104]. Given the high stakes, the imperative for new approaches is greater than ever. Flexible and reflective treatment protocols will be our only chance to lower death rates and eventually contribute to the control of this pandemic. We therefore agree that COVID-19 is indeed (also) a true vascular and endothelial disease. We also suggest that a “single-drug approach” would be insufficient to address the COVID-19 coagulopathy. We suggest that the treatment of COVID-19 patients, might be based on results from point-of-care analyses such as the TEG^®^, as well as detailed analysis of microclot presence as well as platelet activation, that shows the physiological status of the hematological and coagulation system in real-time. We do realize that a microclot ad platelet grading system is using fluorescence microscopy is not the most user-friendly techniques, and quantification methods would provide better standardization. However, currently there are no such methods available. Developing more robust and quantifiable methods might most definitely be investigated further. We conclude that, because of the overwhelming alteration of clotting, the outcome of any trial evaluating an any single anticoagulant, including thrombolytic drugs, would be negative. A multi-pronged approach, guided by TEG^®^ and platelet mapping, would be required to maintain normal clotting physiology in severe COVID-19 disease.

## Figures and Tables

**Figure 1 jcm-10-05381-f001:**
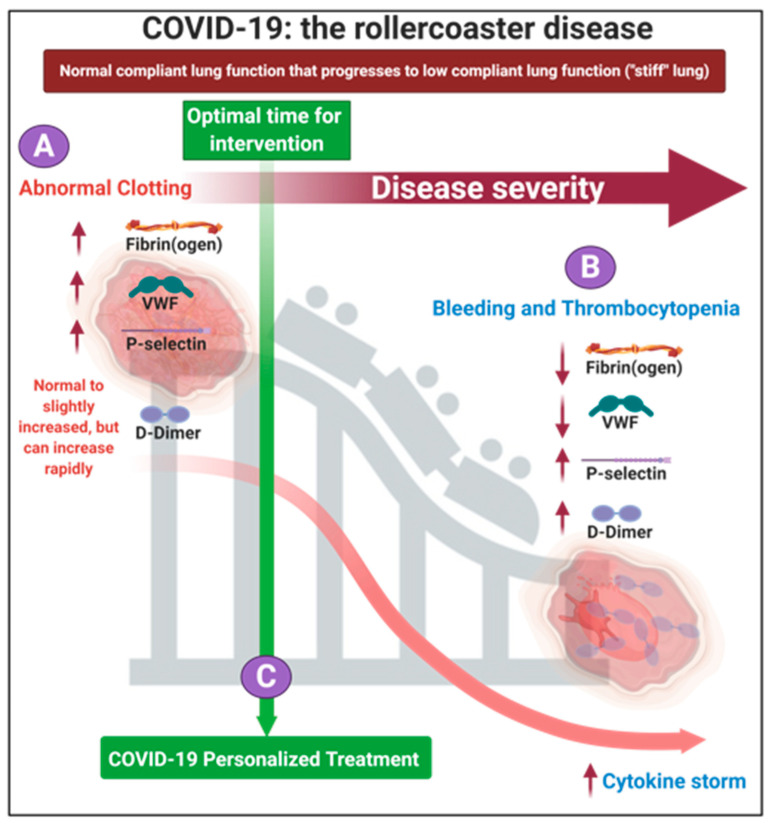
The rollercoaster vascular pathology in acute respiratory syndrome coronavirus 2 (COVID-19) (adapted from [41]). We focus on fibrin(ogen), D-Dimer, P-selectin and von Willebrand Factor dysregulation, resulting in endothelial, erythrocyte and platelet dysfunction. (**A**) Early on in the disease dysregulation in clotting proteins and circulating biomarkers may occur and is suggestive of hypercoagulation. (**B**) The disease may progress to bleeding and thrombocytopenia. (**C**) We suggest that each patient could be treated using a personalized medicine approach in the early stages of the disease. Image created with BioRender (https://biorender.com/ (accessed on 15 October 2021)).

**Figure 2 jcm-10-05381-f002:**
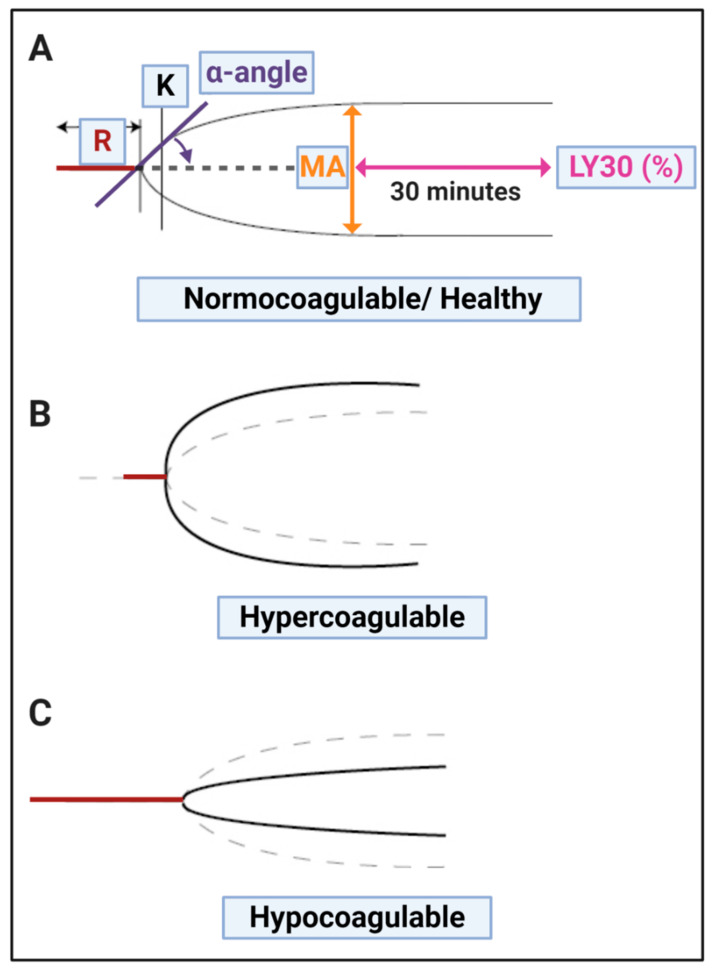
TEG^®^ traces with the various parameters discussed in Table 2, visualized. (**A**) Healthy (normocoagulable) trace; (**B**) Hypercoagulable trace and (**C**) Hypocoagulable trace. Image created with BioRender (https://biorender.com/ (accessed on 15 October 2021)).

**Figure 3 jcm-10-05381-f003:**
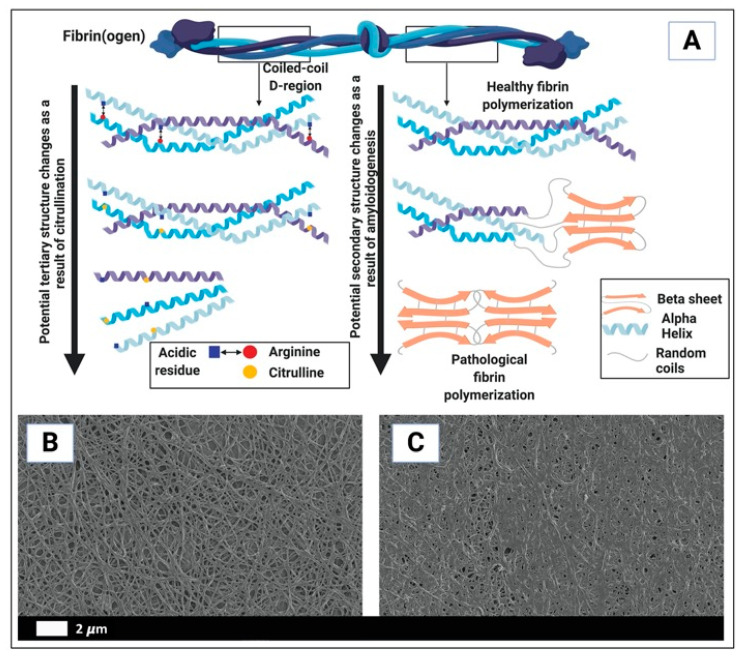
Differences in healthy and pathological clotting biochemistry [76]. (**A**) The uncoiling of the fibrin(ogen) protein (in part) resulting in whole blood and plasma hypercoagulability. (**B**,**C**) are examples of scanning electron microscopy micrographs of (**B**) polymerized fibrin cross-linking fibrin clot from a healthy individual (created with platelet poor plasma with added thrombin); (**C**) fibrin clot from a diabetes individual (created with platelet poor plasma with added thrombin) ((**B,C**) reprinted from [76]). Image in (**A**) created with BioRender (https://biorender.com/ (accessed on 15 October 2021)).

**Figure 4 jcm-10-05381-f004:**
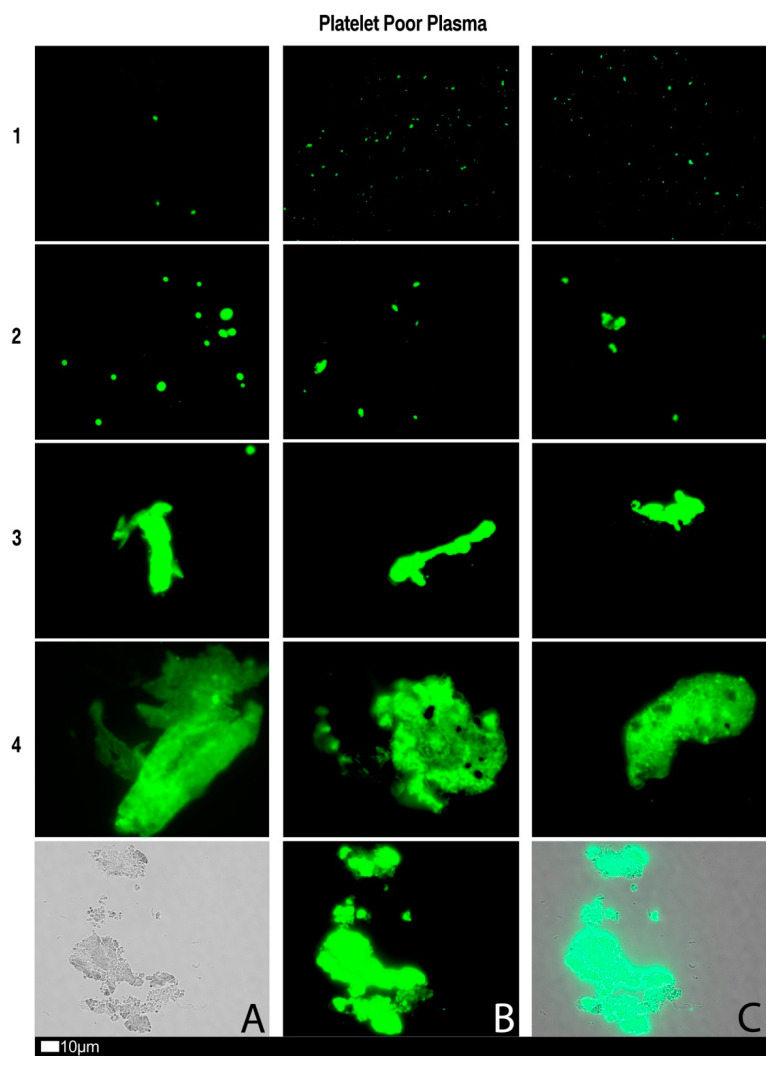
Fluorescence microscopy showing microclots in platelet poor plasma (PPP) with representative examples of the different stages of different stages of microclot formation. Stage 1 shows minimal microclot formation in healthy/control PPP which progresses to the presence of the severe microclotting Stage 4. Stage 1 to 4 is also used as a numerical scoring system, where a score of 1 is given for minimal microclot formation and a score of 4 is given for significant and widespread microclot formation. Bottom row represents examples of stage 4 microclots using (**A**) bright-field microscopy, (**B**) fluorescence microscopy, and (**C**) an overlay of fluorescence and bright-field microscopy.

**Figure 5 jcm-10-05381-f005:**
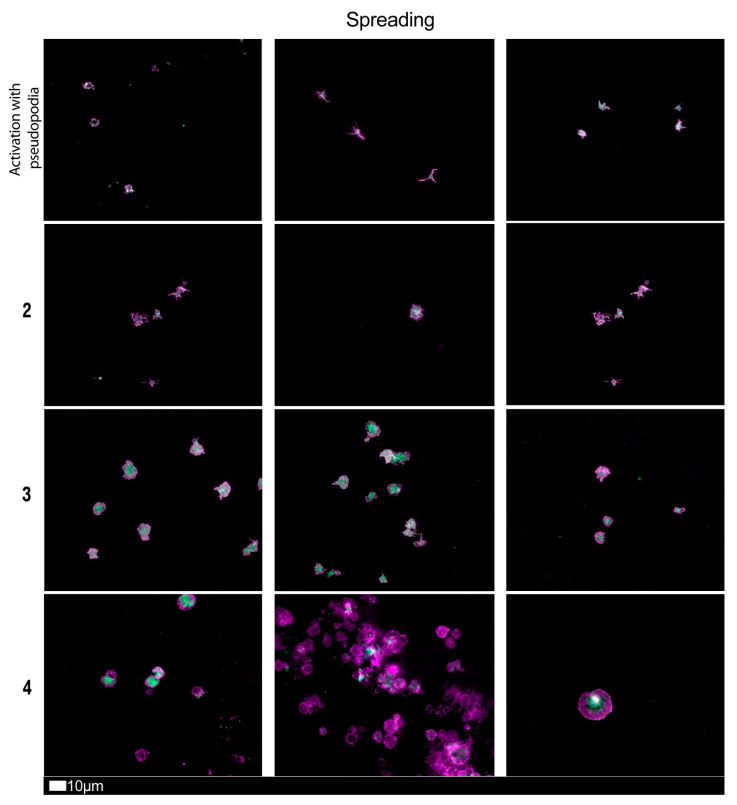
Fluorescence microscopy examples of the different stages of platelet activation and spreading. From healthy/control samples, with minimally activated platelets, seen as small round platelets with a few pseudopodia visible due to contact activation as seen in Stage 1, that progresses to the egg-shaped platelets, indicative of spreading and the beginning of clumping, as seen in Stage 4. Stage 1 to 4 is also used as a numerical scoring system, where a score of 1 is given for slight activation and pseudopodia formations, seen in a healthy individual, and a score of 4 is given for spreading.

**Figure 6 jcm-10-05381-f006:**
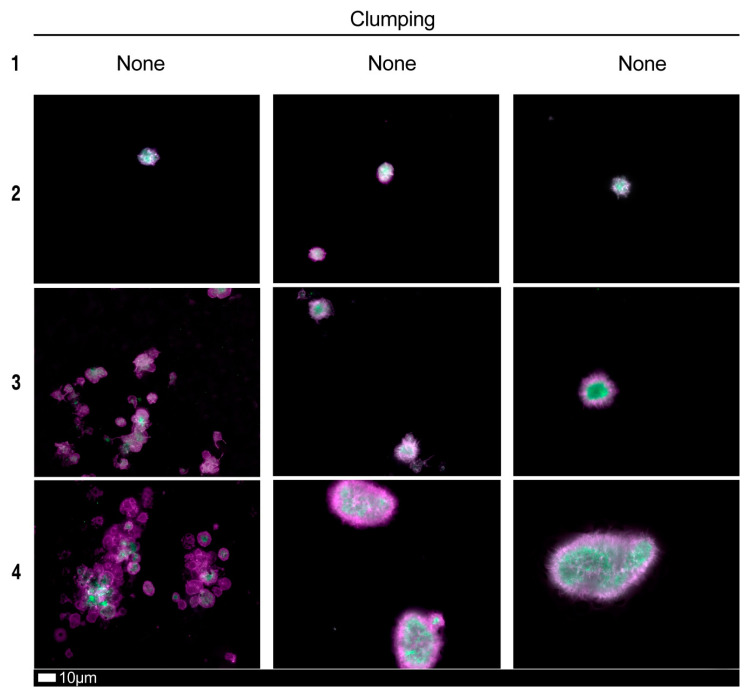
Fluorescence microscopy examples of the different stages of platelet clumping. With no clumping occurring in the healthy/control samples in Stage 1 (no figures shown), progressing to severe clumping of platelets as seen in Stage 4. Stage 1 to 4 is also used as a numerical scoring system, where a score of 1 is given for no clumping and a score of 4 is given for severe clumping.

**Table 1 jcm-10-05381-t001:** Dysregulation of circulating biomarkers P-selectin, von Willebrand Factor, fibrin(ogen) and D-dimer in COVID-19. See Figure 1 for levels during COVID-19.

Circulating Biomarkers	Selected References
P-selectin	[7,51]
Fibrin(ogen) and D-dimer	[38,42,45,47,52,53,54,55,56,57,58]
Von Willebrand Factor	[17,36,59,60]

**Table 2 jcm-10-05381-t002:** (**A**) Thromboelastography^®^ (TEG^®^) clot parameters for whole blood and (**B**) PFA-200 platelet parameters.

(**A**) **Thromboelastography®**		
**TEG® Parameters**	**Explanation**	
R value: reaction time measured in minutes	Time of latency from start of test to initial fibrin formation (amplitude of 2 mm); i.e., initiation time.	
K: kinetics measured in minutes	Time taken to achieve a certain level of clot strength (amplitude of 20 mm); i.e., amplification.	
A (Alpha): Angle (slope between the traces represented by R and K) Angle is measured in degrees	The angle measures the speed at which fibrin build up and cross linking takes place, hence assesses the rate of clot formation, i.e., thrombin burst.	
MA: Maximal Amplitude measured in mm	Maximum clot size: it reflects the ultimate strength of the fibrin clot, i.e., overall stability of the clot. The larger the MA the more hypercoagulable the clot.	
Maximum rate of thrombus generation (MRTG) measured in Dyn.cm^−2^.s^−1^	The maximum velocity of clot growth observed or maximum rate of thrombus generation using G, where G is the elastic modulus strength of the thrombus in dynes per cm^−2^.	
Time to maximum rate of thrombusgeneration (TMRTG) measured in minutes	The time interval observed before the maximum speed of the clot growth.	
Total thrombus generation (TTG) measured in Dyn.cm^−2^	The clot strength: the amount of total resistance (to movement of the cup and pin) generated during clot formation. This is the total area under the velocity curve during clot growth, representing the amount of clot strength generated during clot growth.	
Lysis at 30 min (LY30) measured in %	The LY30 parameter (measured in %) is recorded at 30 min, measured from the point where the maximum amplitude (MA) of the clot is reached.	
G value measured in Dyn.sec	G-value is a log-derivation of the MA and is meant to also represent the clot strength Elevated G-value is associated with a hypercoagulable state and therefore increases the risk for venous thromboembolic disease.	
(**B**) **PFA-200 Platelet Test Interpretation:**		
Citrated whole blood is aspirated at high shear rates through disposable cartridges. These cartridges contain an aperture within a membrane coated agonist. The agonist cartridges are Col/EPI, Col/ADP and P2Y and they report data in closure time. The PFA-200 test induces platelet adhesion, activation and aggregation using the three cartridges. Closure times increase progressively as the platelet counts falls below 100 × 109/L.		
Agonist cartridges [66]	Test principle	Closure time interpretations: measured in seconds
Collagen and epinephrine (Col/EPI): This cartridge has a collagen (2 μg equine type I) and epinephrine (10 μg)-coated membrane (C/Epi).	Co-stimulation by shear stress, collagen, epinephrine. Gives an indication of effectiveness of aspirin and GP IIβ/IIIα inhibitor dosage.	Col/EPI closure time is 82–150 s with a value >150 s regarded as prolonged.
Collagen and ADP (Col/ADP): This cartridge has a collagen (2 μg equine type I) and adenosine-diphosphate (50 μg)-coated membrane (Col/ADP).	Co-stimulation by shear stress, collagen, ADP. Gives an indication of effectiveness of aspirin and clopidogrel and GP IIβ/IIIα inhibitor dosage.	Col/ADP closure time is 62–100 s with a value >100 s regarded as prolonged.
P2Y: This cartridge has a prostaglandin E1 (5 ng) and ADP (20 μg)-coated membrane.	Co-stimulation by shear stress, ADP, PGE1 and Ca2+. Gives an indication of effectiveness of clopidogrel and GP IIβ/IIIα inhibitor dosage	Shortened PFA P2Y closure times >106 s are viewed as prolonged.

**Table 3 jcm-10-05381-t003:** Platelet activation criteria showing level of spreading, as well as clumping in the hematocrit sample.

Score	Spreading	Score	Clumping
1	Activation with pseudopodia	1	None
2	Mild	2	Mild
3	Moderate	3	Moderate
4	Severe	4	Severe

**Table 4 jcm-10-05381-t004:** Microclot criteria to determine the amount of microclots in the platelet poor plasma sample.

Score	Presence of Microclots in Platelet Poor Plasma
1	Very few areas of plasma protein misfolding (≤1 µm) visible with a few ≤10 µm microclots
2	Very few areas of plasma protein misfolding (≤1 µm) visible with scattered/mild ≤10 µm microclots
3	Moderate areas of plasma protein misfolding visible as microclots ≥15 µm
4	Severe areas of plasma protein misfolding visible as large microclots

**Table 5 jcm-10-05381-t005:** Overall microclot and platelet activation score results.

Control/Healthy	Mild	Moderate	Severe
=3	4–7	8–10	11–12
Platelets + PPP scores =			

**Table 6 jcm-10-05381-t006:** A suggested prognostic indicator based on a points system.

Prognostic Indicator			
Points assigned	0	1	2
Age (years)	≤44	45–64	≥65
Effort intolerance above baseline	No	Yes	
Hypoxemia, O_2_ saturation	>95	92–95	<92
Chest X-ray/CT scan	Normal	≤1 quad	≥1 quad
Obesity (BMI)	<26	26–36	>36
Co-morbidities: 1 point for each comorbid condition	Type 2 Diabetes Mellitus, coronary artery disease (CAD), non-atrial fibrillation (AF) stroke, smoking, deep vein thrombosis (DVT), hyperparathyroidism (HPT), chronic kidney disease (CKD)		
TEG: MA	<69	69–75	>75
TEG: G-score	<10	10–15	>15
TEG: Ly-30	>0	None (1 point)	
Score			
Low Risk	Moderate Risk	High Risk	
0–3	4–10	≥11	

## Data Availability

Data available at https://1drv.ms/u/s!AgoCOmY3bkKHi6F4DXeX2CZioXtX4A?e=A52IdL.

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
