# Peer review of "TEG®, Microclot and Platelet Mapping for Guiding Early Management of Severe COVID-19 Coagulopathy"

_jcm, 2021, doi:10.3390/jcm10225381_

Round 1
Reviewer 1 Report
This is an interesting in-depth review on coagulopathy in severe COVID-19 including the discussion of therapeutic startegies and - mainly - use of advance point-of-care diagnostics guiding therapy.
I have only a few comments:
Title: I suggest to avoid "should" sentence as the title of a review. Maybe better "TEG, Microclot and platalet mapping for guiding early management of severe COVID-19 coagulopathy".
Many clinicians like clear and short guidelines for management. A figure or table to link POCT results to possible therapeutic consequences maybe could help.
Author Response
Dear reviewer 1, please find out responses
Reviewer 1
- Title: I suggest to avoid "should" sentence as the title of a review. Maybe better "OK". changed to:
TEG®, Microclot and Platelet Mapping for Guiding Early Management of Severe COVID-19 Coagulopathy
- Many clinicians like clear and short guidelines for management. A figure or table to link POCT results to possible therapeutic consequences maybe could help.
We added table 4 and the following sentences just before the conclusion.
A prognostic indicator score is also suggested to determine risk of developing severe disease (see Table 4). This score indicator system could allow the clinician to allocate points for various parameters, including age, effort intolerance, Hypoxemia, O2 saturation, chest Z-ray and/or CT scan carotid intima-media thickness, and other co-morbidities. In addition, a scoring based on parameters from the point-of-care TEG is also suggested.
Table 4: Prognostic indicator based on a points system. (please see in the paper itself - it did not paste well here)
Table 4: Prognostic indicator based on a points system.
|
Prognostic indicator |
|||||
|
Points assigned |
0 |
1 |
2 |
||
|
Age (years) |
≤ 44 |
45-64 |
≥65 |
||
|
Effort intolerance above baseline |
No |
Yes |
|
||
|
Hypoxemia, O2 saturation |
> 95 |
92-95 |
< 92 |
||
|
Chest X-ray/CT scan |
Normal |
≤ 1 quad |
≥ 1quad |
||
|
Obesity (BMI) |
< 26 |
26-36 |
> 36 |
||
|
Co-morbidities: 1 point for each comorbid condition |
Type 2 Diabetes Mellitus, coronary artery disease (CAD), non-atrial fibrillation (AF) stroke, smoking, deep vein thrombosis (DVT), Hyperparathyroidism (HPT), chronic kidney disease (CKD) |
||||
|
TEG: MA |
< 69 |
69-75 |
> 75 |
||
|
TEG: G-score |
< 10 |
10-15 |
> 15 |
||
|
TEG: Ly-30 |
> 0 |
None (1 point) |
|||
|
Score |
|||||
|
Low Risk |
Moderate Risk |
High Risk |
|||
|
0-3 |
4-10 |
≥ 11 |
|||
Reviewer 2 Report
The manuscript entitled “Early Management of Severe COVOD-19 Coagulopathy sholud be Guided by TEG®, Microclot and Platelet Mapping” is a review that focuses on a interesting topic regarding the severe COVID-19 coagulopathy in terms of pathogenesis and detection of early diagnostic coagulation tests.
I think that the authors would discuss and expand the argument regarding the limitations of the studied coagulation assays in the COVID-19 such as D-dimer (DD), fibrinogen (Fib), standard coagulation laboratory assays such as prothrombin time (PT) and activated partial thromboplastin time (APTT), and platelet-secreted proteins such as b-Thromboglobulin (b-TG) and Platelet Factor-4 (PF4).
Infact, it is reported that DD has low specificity and elevated levels may be related with other conditions. The fibrinogen concentration is a static measure and does not provide informations about its functionality. PT and APTT are a measure of the plasma clotting activity that ignore other components of the coagulation such as the platelets and the fibrinolysis. The plasma levels of PF4 are affected by a rapid remove from circulation through the binding to endothelial cells or by the mobilization from vascular endothelium on heparin infusion. b-TG levels are not affected by these interference factors. Therefore, I disagree with the statement “…the analysis of platelet activation is extremely difficult,…” P11 3.2. Methods used to prepare platelets L 2.
These comments are important for the readers and scientific community.
I invite the authors to read these papers (Kabrhel C et al Acad Emerg Med 2010; Chaudhary R et al Thromb Haemost 2020; Kaplan KL et al Blood 1981; Busch C et al Thromb Res 1980; Dawes J et al Thromb Res 1978; Liao D et al Lancet Haematol 2020; Amgalan A et al J Thromb Haemost 2020).
I have difficulty reading this manuscript because the concepts are expressed in a confused way and the reading is not very fluent.
I think that this manuscript is not suitable for publication in its actual version.
Author Response
Dear reviewer 2 thank you for your valuable comments! Please see our responses in the color version of the manuscript in red
Reviewer 2
The manuscript entitled “Early Management of Severe COVOD-19 Coagulopathy sholud be Guided by TEG®, Microclot and Platelet Mapping” is a review that focuses on a interesting topic regarding the severe COVID-19 coagulopathy in terms of pathogenesis and detection of early diagnostic coagulation tests. Thank you
I think that the authors would discuss and expand the argument regarding the limitations of the studied coagulation assays in the COVID-19 such as D-dimer (DD), fibrinogen (Fib), standard coagulation laboratory assays such as prothrombin time (PT) and activated partial thromboplastin time (APTT), and platelet-secreted proteins such as b-Thromboglobulin (b-TG) and Platelet Factor-4 (PF4). The plasma levels of PF4 are affected by a rapid remove from circulation through the binding to endothelial cells or by the mobilization from vascular endothelium on heparin infusion. b-TG levels are not affected by these interference factors.
Thank you for the suggestions and the very significant references. We read them with care and added them into various sections of the paper – in red.
We added the following paragraph with references:
Typical laboratory pathology tests are usually done on plasma (after the cellular material has been removed that includes the platelets). In these tests the platelets are literally discarded and therefore ignored. Well-known coagulation tests like the prothrombin time (PT) and partial thromboplastin time (PTT), has been shown not to give a true reflection of the hypercoagulable state in acute COVID patients [78], as these parameters ignore other components of the coagulation such as the platelet function and fibrinolysis [31]. On the other hand, whole blood viscoelastic analysis can be rapidly performed by TEG or ROTEM, as these techniques measure the whole blood capability to make and sustain clot formation [31]. Variability between study protocols and results should be kept to the minimum by standardized protocols to investigate the utility of TEG/ROTEM in assessing risk for thrombosis and bleeding [79].
Infact, it is reported that DD has low specificity and elevated levels may be related with other conditions. The fibrinogen concentration is a static measure and does not provide informations about its functionality. PT and APTT are a measure of the plasma clotting activity that ignore other components of the coagulation such as the platelets and the fibrinolysis.
Also see line 157 to 164, and 281 to 283 and table 1 where D-dimer levels are discussed.
Yes we agree! We added these sentences to reiterate:
D-dimer might also have low specificity and elevated levels may be related with other conditions.
During the progression of COVID-19, the circulating biomarkers P-selectin, VWF, fibrin(ogen) and D-dimer may either be within healthy levels, upregulated or eventually depleted [21, 22]. In COVID-19 patients, dysregulation, has been noted in each of them and this may lead to the extensive endotheliopathy noted in COVID-19 patients [8, 48, 49] (see Table 1). Endotheliopathy could give rise to hypercoagulation by alteration in the levels of different factors such as VWF [10]. Fibrinogen concentration is also a static measure and does not provide information about functionality. D-dimer might also have low specificity and elevated levels may be related with other conditions.
Agree see paragraph we added.
Therefore, I disagree with the statement “…the analysis of platelet activation is extremely difficult,…” P11 3.2. Methods used to prepare platelets L 2.
I am not sure how the address this statement of platelet activation analysis, as platelets really do activate easily. The sentence now reads:
The analysis of platelet activation might be extremely difficult, as platelets are easily activated.
These comments are important for the readers and scientific community.
I invite the authors to read these papers (Kabrhel C et al Acad Emerg Med 2010; Chaudhary R et al Thromb Haemost 2020; Kaplan KL et al Blood 1981; Busch C et al Thromb Res 1980; Dawes J et al Thromb Res 1978; Liao D et al Lancet Haematol 2020; Amgalan A et al J Thromb Haemost 2020).
Thank you for these references! We added it the various paragraphs
Also this sentence in the conclusion:
Beta-thromboglobulin and platelet factor 4 may also be used for detecting increased platelet activation in vivo [99, 100]. In 1981, Kaplin already pointed out that the measurement of plasma levels of beta-thromboglobulin and platelet factor 4 can be is useful [101].
In section 1.1 about DIC:
It was also noted that sepsis-induced coagulopathy and the International Society of Thrombosis and Hemostasis(ISTH) overt DIC scores (assessed in 12 patients who survived and eight patients who died), increased over time in patients who died. The onset of sepsis-induced coagulopathy was typically before overt DIC [37] .
Reviewer 3 Report
This work is a comprehensive review of the coagulopathy of COVID-19 infection, and cites many key clinical studies and key references. In particular, the description on p. 6 of the parameters of TEG and their meaning is excellent. The authors' contribution to a new assessment consists of two points: 1) the importance of using TEG to control and monitor anticoagulation, and 2) the importance of including assays of platelet function. The authors describe interesting assays described as a fluorescent microclot assessment, and measures of platelet spreading and clumping, using known and accepted antibodies known to recognize platelet surface P-selectin and the active conformation of platelet surface GPIIb/IIa. However, the platelet assays are conducted in static, platelet-rich plasma (PPP) and would benefit from having some key element of red blood cells and blood rheology present, as is intrinsically the case with TEG. The mechanism whereby thioflavin identifies microclots is not presented. Does thioflavin bind misfolded fibrin specifically? Is it incorporated into such fibrin?
No assessment of COVID-19 coagulopathy is given for the pediatric population, for which COVID-19 mortality and morbidity is significantly less: only 2-3 deaths per 1,000,000 children, an incidence of MIS-C of about one in 30000, and an incidence of thrombotic complications of about 2-3 per 100 cases of clinically significant COVID-19. Why is this so? What does this say about mechanism or risk factors for the coagulopathy?
p. 9. How dose heparin interfere with VWF, platelet activation?
pp. 10 (bottom) and 11 (top). There are two incomplete sentences.
Author Response
Dear reviewer 3, thank you for your valuable comments. Please see our responses in the colour version of the paper in blue
This work is a comprehensive review of the coagulopathy of COVID-19 infection, and cites many key clinical studies and key references. In particular, the description on p. 6 of the parameters of TEG and their meaning is excellent. The authors' contribution to a new assessment consists of two points: 1) the importance of using TEG to control and monitor anticoagulation, and 2) the importance of including assays of platelet function. The authors describe interesting assays described as a fluorescent microclot assessment, and measures of platelet spreading and clumping, using known and accepted antibodies known to recognize platelet surface P-selectin and the active conformation of platelet surface GPIIb/IIa.
However, the platelet assays are conducted in static, platelet-rich plasma (PPP) and would benefit from having some key element of red blood cells and blood rheology present, as is intrinsically the case with TEG.
Thank you - we added this following sentence:
In addition, in future the roles of erythrocytes and blood rheology should also be further investigated [104].
The mechanism whereby thioflavin identifies microclots is not presented. Does thioflavin bind misfolded fibrin specifically? Is it incorporated into such fibrin?
We added the following sentence to clarify ThT binding:
ThT binds to open hydrophobic areas on damaged protein [58, 82].
No assessment of COVID-19 coagulopathy is given for the pediatric population, for which COVID-19 mortality and morbidity is significantly less: only 2-3 deaths per 1,000,000 children, an incidence of MIS-C of about one in 30000, and an incidence of thrombotic complications of about 2-3 per 100 cases of clinically significant COVID-19. Why is this so? What does this say about mechanism or risk factors for the coagulopathy?
We have not looked at the paediatric population and this is indeed an interesting question.
p. 9. How dose heparin interfere with VWF, platelet activation?
This sentence was changed to now read:
Heparin assists in the prevention of thrombotic events, by interacting with anti-thrombin III.
pp. 10 (bottom) and 11 (top). There are two incomplete sentences.
Thank you we changed the sentences now to read;
In this trial, 3331 patients were screened and 615 were randomly allocated (311 [50%] to the therapeutic anticoagulation group and 304 [50%] to the prophylactic anticoagulation group. In the study, 576 (94%) were clinically stable and 39 (6%) were clinically unstable.
Round 2
Reviewer 2 Report
The authors responded satisfactorily to the comments.
Therefore, the manuscript is suitable for the publication in its current revised structure.
Author Response
Thank you! Resia and co-authors
Reviewer 3 Report
I would write "ThT binds to open hydrophobic regions...."
Author Response
This sentence was changed to read; ThT binds to open hydrophobic regions on damaged protein..
Thank you
Resia and co-authors